# Melt volume at Atlantic volcanic rifted margins controlled by depth-dependent extension and mantle temperature

Gang Lu [1]✉ & Ritske S. Huismans[1]

Breakup volcanism along rifted passive margins is highly variable in time and space. The factors controlling magmatic activity during continental rifting and breakup are not resolved and controversial. Here we use numerical models to investigate melt generation at rifted margins with contrasting rifting styles corresponding to those observed in natural systems. Our results demonstrate a surprising correlation of enhanced magmatism with margin width. This relationship is explained by depth-dependent extension, during which the lithospheric mantle ruptures earlier than the crust, and is confirmed by a semi-analytical prediction of melt volume over margin width. The results presented here show that the effect of increased mantle temperature at wide volcanic margins is likely over-estimated, and demonstrate that the large volumes of magmatism at volcanic rifted margin can be explained by depth-dependent extension and very moderate excess mantle potential temperature in the order of 50–80 °C, significantly smaller than previously suggested.

[1] Department of Earth Science, Bergen University, Bergen, Norway. ✉email: gang.lu@geo.uib.no

Mantle melting during the formation of mid oceanic ridges is relatively well understood and thought to be mostly a function of mantle potential temperature and spreading rate[1,2]. Decompression melting at standard mantle potential temperature and full spreading rates larger than 1.5 cm/year leads to accretion of 4–8 km of magmatic crust, consistent with uniform global oceanic crustal thickness away from hotspots[3,4]. However, the processes controlling the variation of magmatism at rifted margins are not well understood and a source of controversy[5–10]. Rifted margins in terms of the thickness of early oceanic crust can to first order be characterised with three magmatic modes (Fig. 1). (1) Margins with a sharp transition from the continent-ocean boundary (COB) to normal thickness (4–8 km) magmatic oceanic crust[3,4] can be termed normal-magmatic (Mode 1). (2) Margins where magmatic productivity exceeds that expected from decompression melting at normal mantle temperature, expressed in high volumes of extruded volcanics deposited as seaward-dipping sequences (SDRs), over-thickened intruded continental and oceanic crust and regions of magmatic underplating[11] can be considered excess-magmatic margins (Mode 2). (3) Magma-poor (a-magmatic) margins (Mode 3) have little syn-rift magmatism, in some cases exhibiting a broad zone of exhumed mantle with little to no magmatism at the sea floor preceding formation of mature oceanic crust[12]. While a variety of mechanisms, including low mantle potential temperature[13], low spreading rate[3] and counterflow of depleted lithospheric mantle[14,15], have been suggested as an explanation for the absence of magmatism on magma-poor margins, what controls the volume, distribution and timing of magmatism at normal- to excess-magmatic margins is incompletely understood. The voluminous magmatism at volcanic margins has commonly been explained with mantle plumes, typically with a plume head diameter in the order of 2000 km and excess temperatures ranging 100–200 °C above normal[5,16–18]. However, this interpretation has been challenged by the inferred lack of associated mantle plumes at some volcanic margins such as the US East Coast and NW Australian volcanic margins[6,19]. Moreover, the excess temperature required to produce ultra-thick igneous crust is often in conflict with inferences from geophysical and geochemical analysis[20–22]. Alternative models for voluminous magmatism at volcanic margins include the effects of active upwelling[22,23], rift history[10], small-scale convection[17,24,25] or variation in mantle composition[26,27].

Previous models of melt generation have mostly focused on seeking heterogeneities in temperature or composition of the sub-lithospheric mantle, implicitly assuming simple, uniform lithospheric extension where the crust and mantle lithosphere rupture simultaneously. However, observations have shown that rifted margins rarely experience uniform extension; rather, many margins exhibit complex tectonic styles with depth-dependent extension[28–30]. Narrow margins with coupled deformation in the lithosphere are expected to exhibit early and sharp rupture of both the crust and the mantle lithosphere[14,15]. In contrast at some wide margins, the stretching factor of the crust is significantly smaller than the whole lithosphere[28,29], implying preferential removal of most of the mantle lithosphere. Similar removal of mantle lithosphere is also observed in the Basin and Range wide rift system, where syn-extensional magmatism over a wide range has been identified[31]. These contrasting styles of rifting are to first order controlled by crustal rheology[14,15,32,33].

Here we show that these tectonic rifting styles lead to highly contrasting magmatic outputs during passive margin formation. While narrow rifts are expected to produce mature mid-ocean ridge spreading following early crust and mantle lithosphere rupture, resulting in standard oceanic crust thickness at the COB, wide rifts are expected to lead to significant melt production beneath moderately extended crust before lithospheric rupture. By combining forward models and published observations, we provide a new conceptual and quantitative framework explaining the volume of decompression melting accreted to rifted passive margins as a function of margin width and mantle potential temperature.

## Results

**Numerical model setup.** We use thermo-mechanically coupled finite-element models for the solution of plane-strain, incompressible viscous-plastic creeping flows to investigate extension of a layered lithosphere with frictional-plastic and thermally activated power-law viscous rheologies and consequences for melt generation during rifted margin formation (see "Method" for details on model description, Supplementary Fig. 1 for model setup, and Supplementary Table 1 for model parameters). The model consists of a horizontally layered crust, lithospheric mantle and sub-lithospheric mantle. Initial temperature is laterally homogeneous and the sub-lithospheric mantle has a constant

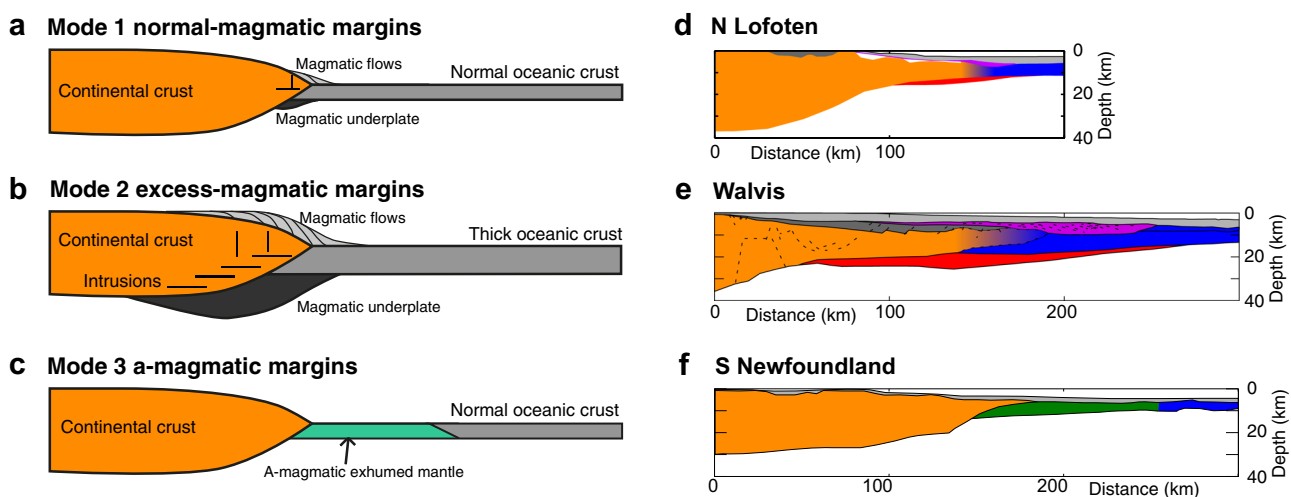

**Fig. 1 Magmatic modes of rifted margins.** Classification of rifted margins in terms of their magmatic modes: **a** normal-magmatic (Mode 1), **b** excess-magmatic (Mode 2) and **c** a-magmatic (Mode 3). Natural examples for the three magmatic modes: **d** N. Lofoten margin[47], **e** Namibian Walvis margin[67] and **f** Newfoundland margin[56].

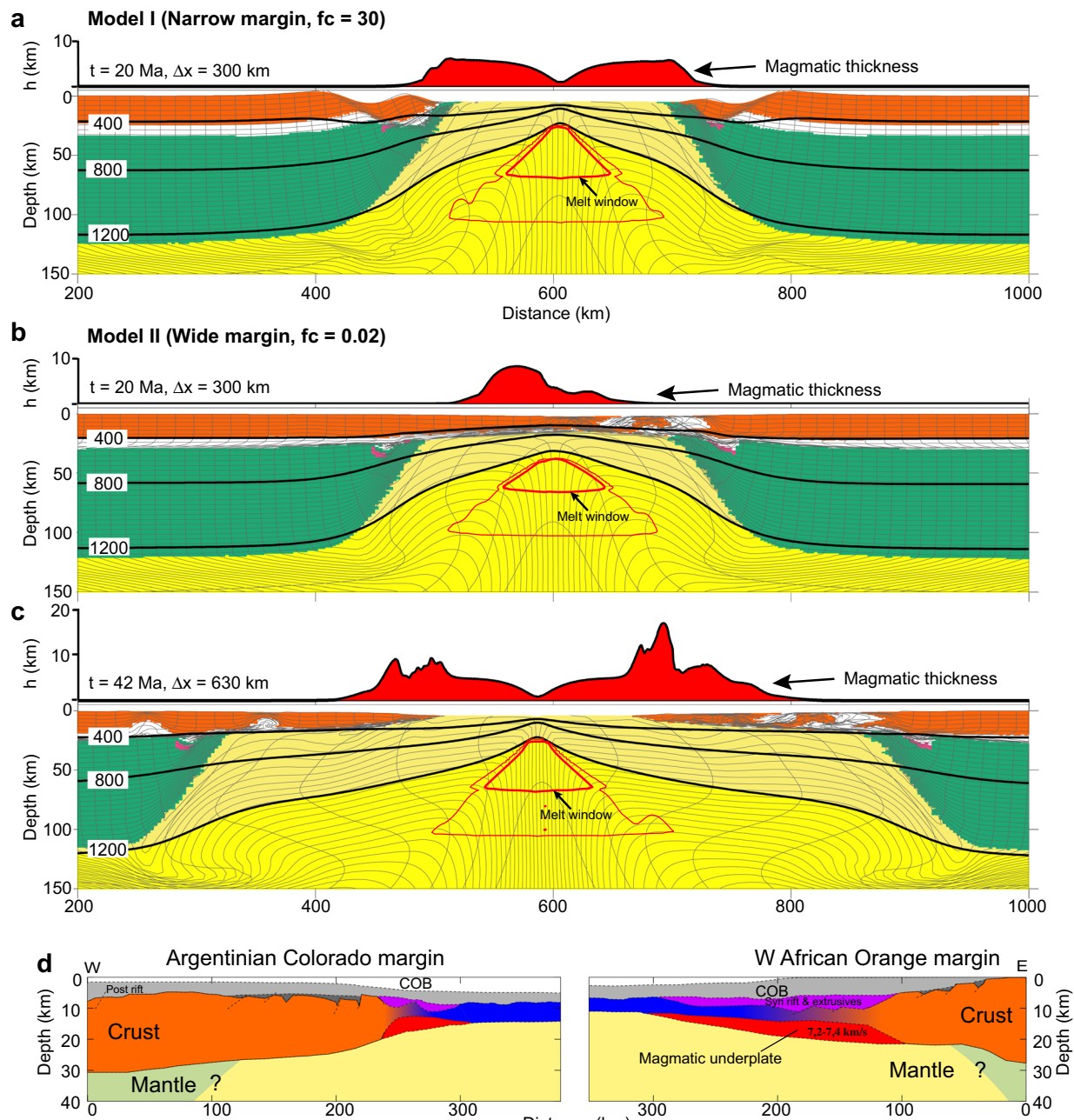

**Fig. 2 Model evolution of contrasting rifting styles. a** Model I with strong crust ($f_c = 30$). Bottom: composition overlain with contours of isotherms (black lines) in degree Celsius and incremental melt fraction (red lines). The thick red lines show melt windows with major decompression melting. Phase colours: upper crust, orange; lower crust, white; continental mantle lithosphere, green; asthenosphere, yellow; and oceanic lithosphere, pale yellow. Top: predicted magmatic thickness. *t* time since the onset of extension, Ma millions of years; $\Delta x$, extension at full velocity 1.5 cm/year. **b**, **c** Model II with weak crust ($f_c = 0.02$). Note the earlier rupture of mantle lithosphere than crust and enhanced magmatic production in the distal margin. **d** Cross sections of wide Southern South Atlantic conjugate margins[68]. Colouring as in **a**. Also shown are magmatic underplate (red), extrusives (purple), oceanic crust (blue), syn- (dark grey) and post-rift (grey) sediments. COB continent-ocean boundary.

mantle potential temperature ($T_p$). We explore models with varying crustal strength to investigate the role of contrasting styles of rifted margin formation on magmatism. A Wet Quartz flow law is used for the crust[34], which is scaled by a viscosity-scaling factor, $f_c$, to produce stronger or weaker crust. The melt parameterization model follows ref.[24] (see "Methods" for details). Melt parameters are calibrated by comparing predicted igneous crustal thickness with global oceanic crustal thickness[3], with a mantle potential temperature of 1300 °C resulting in on average 6-km thick oceanic crust (Supplementary Fig. 2).

**Volcanic rifted margin models**. Reference Model I (Fig. 2a) with strong crust ($f_c = 30$) and normal mantle potential temperature, $T_p = 1300$ °C, leads to narrow lithospheric breakup. The strong coupling between frictional-plastic upper crust and upper mantle lithosphere promotes narrow rupture of the whole lithosphere. The transition from the COB to normal oceanic crust is within a distance of <30 km, with predicted melt thickness (i.e. igneous crustal thickness) gradually increasing from 0 to ~5.5 km, in the range of normal global oceanic crust thicknesses[3,4]. Model II shows highly contrasting behaviour, with very weak crust ($f_c = 0.02$) allowing for

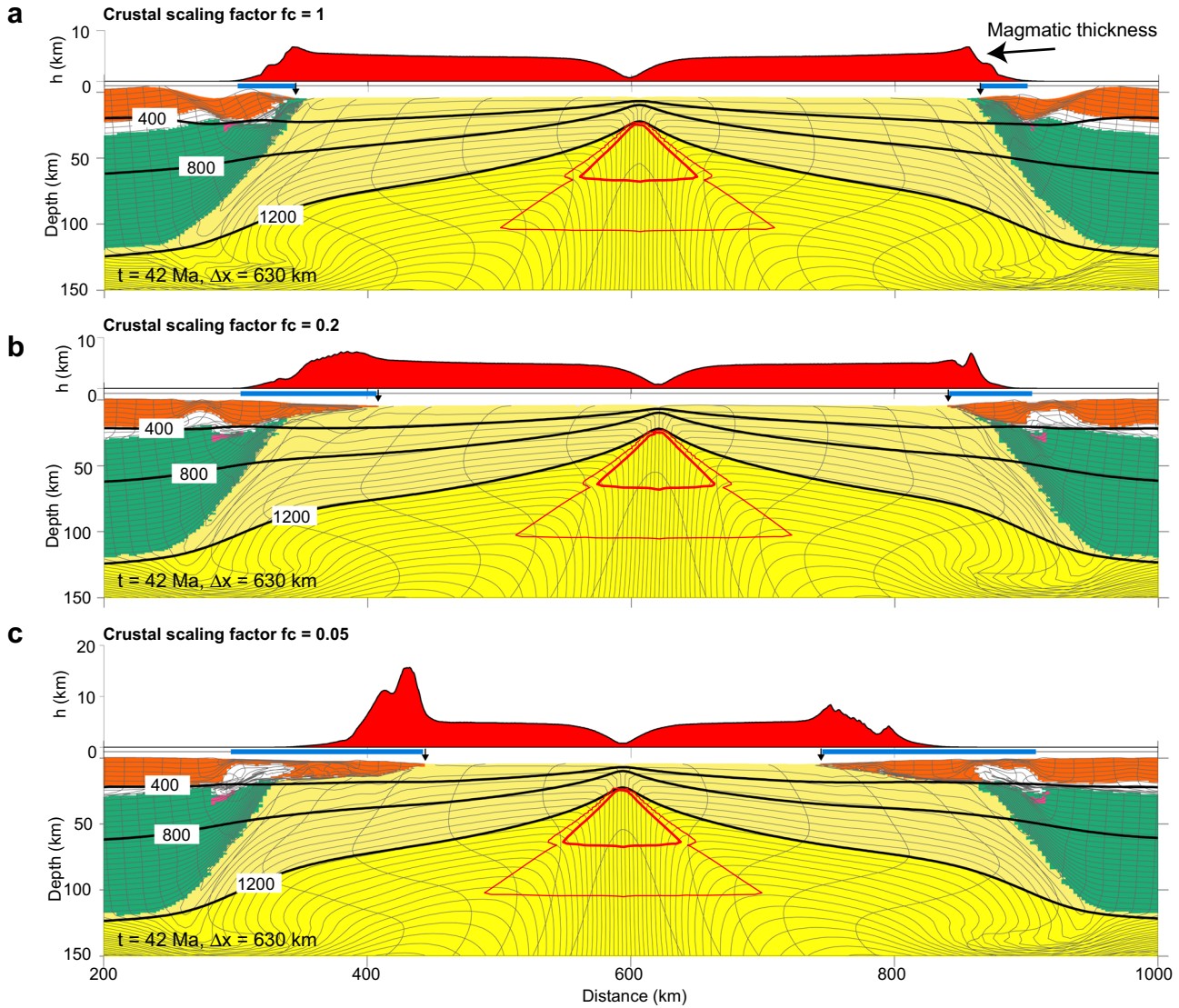

**Fig. 3 Melt production for models with intermediate crustal strength between end-member models I and II. a–c** Snapshots of models with decreasing crustal strength as represented by the Wet Quartz rheology[34] with viscosity-scaling factors ($f_c$) from 1 to 0.05, leading to increasing margin width and melt thickness at the distal margin. All models shown are at the same time and amount of extension as the final stage of Model II in Fig. 2. Black arrows indicate COB. Blue bars indicate margin width.

decoupling of upper crust and mantle lithosphere leading to highly depth-dependent extension, leaving the extended crust in contact with upwelling sub-lithospheric mantle (Fig. 2b, c). Depth-dependent thinning results in distinctly different magmatic productivity, with mantle lithospheric rupture beneath the extending crust allowing for syn-rift decompression melting (Fig. 2b) of the upwelling sub-lithospheric mantle and voluminous magma production accreted to the distal margin (Fig. 2c), with peak melt thickness (~18 km) more than three times thicker as compared to narrow rift Model I (Fig. 2a). The large amount of melt accretion to the distal margin is explained by preferential removal of the mantle lithosphere during depth-dependent extension. Corner flow mantle upwelling following mantle lithosphere rupture is controlled by the far field rate of divergence. As distributed extension in the crust above occurs over a much larger horizontal length scale, the horizontal velocity at which the crust moves is significantly lower compared to the rate of mantle upwelling below and the crust therefore collects more melt as it stays longer above the area of mantle melting. Igneous oceanic crust rapidly decreases to

reference thickness of ~5.5 km following crustal breakup consistent with oceanic crustal thickness for normal mantle temperature. Narrow and wide rift models I and II demonstrate highly contrasting magmatic productivity as a function of margin width and consequently crustal strength. Models with systematic variation of crustal strength intermediate between end-member conditions for narrow and wide margin systems ($f_c = 30$ and $f_c = 0.02$) confirm progressive enhancement of magmatic accretion to the distal margin with increasing margin width (Fig. 3) and demonstrate a quasi-linear correlation between margin width and total magmatic volume (Fig. 4) (see Supplementary Fig. 3 for definition of melt volume and margin width). As asymmetry in both margin width and melt distribution may occur for certain conditions (Supplementary Fig. 3), we have calculated the total melt volume from both conjugate margins in order to minimize the influence of asymmetry. Increasing mantle potential temperature by 80 °C above the reference state leads to a similar quasi-linear correlation between total magmatic volume and margin width but with a larger slope (Fig. 4).

**Semi-analytical scaling law**. The quasi-linear correlation between total melt volume ($V^*$) and margin width ($W$) can be parameterised to first order by $V^* = h_{eff}W + V_0$, where $V_0$ is the intercept on the volume axis and $h_{eff}$ is the slope of the linear curve. As we include melt volume over an initial spreading section of 50 km on each side (Supplementary Fig. 3; also see "Methods"), $V_0$ represents the igneous volume related to steady-state oceanic crustal thickness, $h_{oc}$, over an initial spreading section with a total width of $W_s = 100$ km (e.g. $V_0 = h_{oc}W_s$). $h_{eff}$ represents the average igneous crustal thickness produced during wide rifting. In our models, $h_{eff}$ can be derived semi-analytically based on the characteristics of depth-dependent wide rifting ("Methods," Supplementary Fig. 4), which gives $h_{eff} \approx 0.6h_{oc}$ for $T_p = 1300$ °C. The total melt volume at conjugate margins is thus given by $V^* = 0.6h_{oc}W + 100h_{oc}$. Higher potential temperature leads to increased magmatic productivity[1,35] and consequently to a larger reference oceanic crustal thickness $h_{oc}$ and higher slope of the linear relationship, $h_{eff} = 0.6h_{oc}$. This simple relationship shows that the volume of breakup magmatism is a function of both margin width and potential temperature and compares very well with model results for different margin widths and potential temperatures (Fig. 4).

**Magmatic volume and margin width along Atlantic rifted margins**. We next estimate total volume of magmatic addition and margin width for North, Central and South Atlantic conjugate rifted margins based on published seismic refraction and reflection data. Interpretations of the COB and of magmatic addition based only on seismic reflection data are known to be ambiguous. The extent of continental crust in the transition zone, the location of the COB and the volume of magmatic addition at volcanic margins are often difficult to assess and associated with uncertainty[36]. More reliable determination of the location of the COB and the volume of extruded, intruded and underplated magmatism in the distal margin requires combined analysis of high quality reflection and refraction data, and gravity modelling (e.g. refs. [37,38]). We limit our analyses to sections where both conjugate margins are available in order to account for possible asymmetric distribution of magmatic volumes[22,39] and prioritize conjugate margin sections where both refraction and reflection seismic data are available (Table 1).

Volume of magmatic addition is estimated from three contributions[11,40,41] (Fig. 5a): (1) extrusive magmatism expressed as seaward-dipping reflector sequences ($V_{SDR}$) with P-wave velocities increasing from ~4.0 to ~6 km/s, (2) high-velocity (>7.2 km/s) lower crustal bodies ($V_{LCB}$) interpreted as magmatic underplates at the base of the crust and (3) transitional partially intruded crust ($V_{intrude}$) between SDR and LCB. Following ref. [42], the content of igneous material in each contribution is assumed to be 50 ± 50% for SDR, 10 ± 10% for transitional crust and 100% for LCB. Total melt volume $V^*$ per unit margin length along strike is calculated by summing all contributions from both conjugate margins, together with the additional contribution over the first 50-km oceanic spreading section on each side, $V^* = V_{LCB} + 0.5V_{SDR} + 0.1V_{intrude} + V_{spread}$. Errors in melt volume come principally from uncertainty of portions of igneous material in SDR and transitional intruded crustal volumes, and are calculated as $V_{err} = 0.5V_{SDR} + 0.1V_{intrude}$. Estimated total melt volumes are listed in Table 1 (see Supplementary Table 2 for full list of data sources and uncertainties).

Margin width is defined as the distance between the landward termination of un-thinned continental crust and the

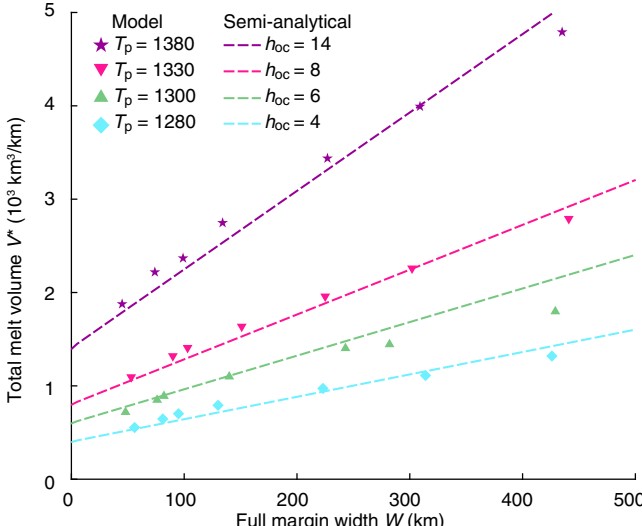

**Fig. 4 Melt volume–margin width correlation.** Model results at mantle potential temperatures of 1280 °C (blue diamonds), 1300 °C (green triangles), 1330 °C (inverted pink triangles) and 1380 °C (magenta stars) all show quasi-linear correlation between margin width and melt volume, respectively. Dashed lines show semi-analytical prediction of melt volume versus width with parameterized oceanic crustal thickness $h_{oc} = 4$ km (blue), 6 km (green), 8 km (pink) and 14 km (magenta).

**Table 1 Magmatic mode classification of North, Central and South Atlantic margins.**

| ID | Name | Width $W$ (km) | Total volume $V^*$ (km²) | $h_{oc}$ (km) | $\bar{h}_{oc}$ (km) | Mode |
|---|---|---|---|---|---|---|
| 1 | Pelotas–Walvis | 316 | $4.36 \times 10^3$ | 15.3 | 15.1 | 2 |
| 2 | Colorado N–Orange N | 306 | $1.83 \times 10^3$ | 7.5 | 6.4 | 1 |
| 3 | Colorado S–Orange S | 253 | $1.84 \times 10^3$ | 7.3 | 7.3 | 1 |
| 4 | Baltimore–Dakhla | 221 | $2.44 \times 10^3$ | 8.5 | 10.5 | 2 |
| 5 | Morocco–Nova Scotia | 312 | $0.31 \times 10^3$ | 3.0 | 1.1 | 3 |
| 6 | Newfoundland N–Iberia N | 190 | $0.09 \times 10^3$ | 2.0 | 0.4 | 3 |
| 7 | Newfoundland S–Iberia S | 272 | $0.00 \times 10^3$ | 0.0 | 0.0 | 3 |
| 8 | SE Greenland–Edoras | 104 | $1.82 \times 10^3$ | 10.7 | 11.2 | 2 |
| 9 | SE Greenland–Hatton Bank | 95 | $2.06 \times 10^3$ | 14.2 | 13.1 | 2 |
| 10 | Jan Mayen–Møre | 162 | $1.71 \times 10^3$ | 6.7 | 8.7 | 2 |
| 11 | NE Greenland–Vøring S | 291 | $4.80 \times 10^3$ | 15.7 | 17.5 | 2 |
| 12 | NE Greenland–Vøring N | 267 | $4.17 \times 10^3$ | 14.0 | 16.0 | 2 |
| 13 | NE Greenland–Lofoten S | 152 | $1.38 \times 10^3$ | 7.4 | 7.2 | 1 |
| 14 | NE Greenland–Lofoten N | 133 | $0.92 \times 10^3$ | 5.8 | 5.1 | 1 |

$h_{oc}$ is measured thickness of early oceanic crust averaged from both sides of conjugate margins. $\bar{h}_{oc}$ is projected thickness of oceanic crust inverted using the semi-analytical scaling law as $\bar{h}_{oc} = V^*/(0.6W + 100)$.

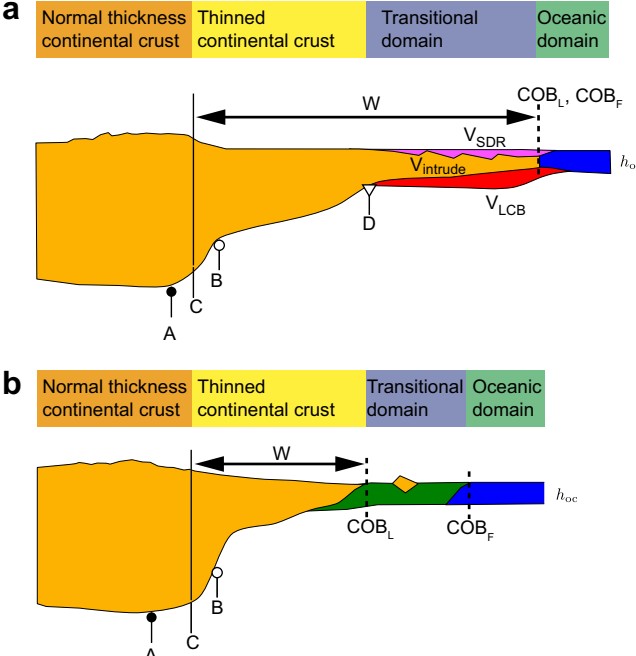

**Fig. 5 Schematic plot for margin width definition. a, b** Margin width measurements for volcanic and magma-poor margins, respectively. Margin width is defined by the distance between the termination of un-thinned continental crust and the continental ocean boundary (COB). The termination of un-thinned continental crust is defined at the mid point (C) of taper zone that starts from the first crustal thinning (point A) and stops at the location where the Moho reaches a depth of 20 km or flattens (point B). $COB_L$: last continental crust; $COB_F$: first oceanic crust.

most distal location of continental crust, e.g. the COB (Fig. 5 and Supplementary Fig. 3). Earlier studies[43,44] suggest that the COB can be defined as either the first identified oceanic crust ($COB_F$) or the ocean ward limit of continental crust ($COB_L$). $COB_F$ and $COB_L$ coincide at volcanic margins, whereas they differ at a-magmatic margins with exhumed mantle. We use therefore the most distal location of continent crust ($COB_L$) as it better captures the degree of crustal stretching of passive margins. Several proxies have been used to define the landward limit of a margin, including the location where the crust reaches a thickness of 25 km (ref. [9]), the location of the onshore topographic maximum[43] and the location of the innermost normal fault[45]. Here we use the termination of un-thinned continental crust as the landward limit of the margin, defined as the mid point of the crustal taper bounded by the location of the first crustal thinning (Fig. 5, point A) and the location where the Moho reaches a depth of 20 km or flattens after rapid thinning (Fig. 5, point B). This approach is similar to that used in earlier studies[9] and provides a simple and robust proxy for the landward limit of rifted margins that can be equally applied to all sections as well as to the numerical models (Supplementary Fig. 3). Uncertainty in margin width results mainly from uncertainty in the location of landward limit of the margin and is greatest if the crustal thinning is gentle. For poly-phase rifted margins, such as the Norwegian margins with intermittent phases of no extension over 50 Ma or longer[45,46], we define margin width based on the last rifting phase that is related with breakup volcanism[16,47] and use the location of most proximal extrusive/underplated magmatism as the landward limit of the last rifting phase.

**Natural rift classification.** The predicted control of margin width and potential temperature on melt volume allows us to characterise natural systems in terms of their magmatic output using observed melt volume and corresponding margin width measured from published North, Central and South Atlantic conjugate rifted margins (Fig. 6, Table 1, Supplementary Table 2 and Supplementary Figs. 5–7). Given the dependency of oceanic crustal thickness on potential temperature (Supplementary Fig. 2), $h_{oc} = h_{oc}(T_p)$, we may divide the melt volume–margin width space into three temperature regimes: (1) a normal-temperature regime (1280–1330 °C) with $h_{oc}$ in the range of 4–8 km, (2) a high-temperature regime (>1330 °C) with $h_{oc} >$ 8 km and (3) a low-temperature regime (<1280 °C) with $h_{oc} <$ 4 km (Fig. 6). Margins that plot in the normal-temperature regime can be considered as normal-magmatic (Mode 1); those in the high-temperature regime as excess-magmatic (Mode 2) margins; and conjugate margin systems in the low-temperature regime as a-magmatic (Mode 3) margins (Fig. 6).

The range of conjugate margin systems that can be understood in terms of normal-magmatic output is unexpected and includes the northern most narrow North Atlantic Lofoten-Greenland margins[47–49], and the very wide conjugate South Atlantic Orange-Colorado margins[5,50]. The Orange-Colorado system, previously interpreted as related to mantle plume activity[5,37], is particularly notable as it is characterised by significant magmatic addition and conjugate margin width in the range 250–300 km. However, the initial oceanic crust thickness of 7.0 km along this conjugate margin[37] is in the range of normal oceanic crust thickness[3]. We show here that the total magmatic volume at this margin is in the range expected for normal-magmatic systems and does not require anomalous high mantle potential temperature. Excess-magmatic conjugate margins span a wide range, with some characterised by only moderately excess activity such as the East US-West African[6,51], the Møre-Jan Mayen[47,52] and the Pelotas-Namibian conjugate margins[50]. Others such as the SE Greenland-UK[8,53] and Vøring-East Greenland[54,55] volcanic margins that are classically interpreted as related to the Iceland plume show clear excess-magmatic volume versus width. However, we show here that these margins require only a moderate potential temperature anomaly in the order of 50–80 °C. The Iberia-Newfoundland and Morocco–Nova Scotia conjugate margins[12,56,57] with intermediate margin width and low melt volume can be typified as a-magmatic systems in agreement with current understanding and have been explained by a range of alternative mechanisms including low mantle potential temperature[13], slow spreading rate[3], compositional inheritance[58], lithospheric counterflow[14] and/or fluid-induced serpentinization[59].

## Discussion

While the results presented here show that voluminous magmatism may be produced from wide rifting at normal mantle temperature, our models do not preclude the involvement of mantle plumes. The effect of enhancing magmatism by margin width occurs for any potential temperature (Fig. 4). At higher potential temperatures, total melt volume increases more rapidly with margin width than at lower temperatures. This implies that, when preferential removal of mantle lithosphere during wide rifting is taken into account, the potential temperature required for the observed amount of magmatism may have been over-estimated. The NE Atlantic large igneous province is a classical example with mantle plume involvement. Seismic studies document igneous crustal thickness of up to ~35 km near the centre of the Iceland hotspot track, and thicknesses ≥15 km extending >1000 km along the margins to the north and south[22,48,60]. Along the SE

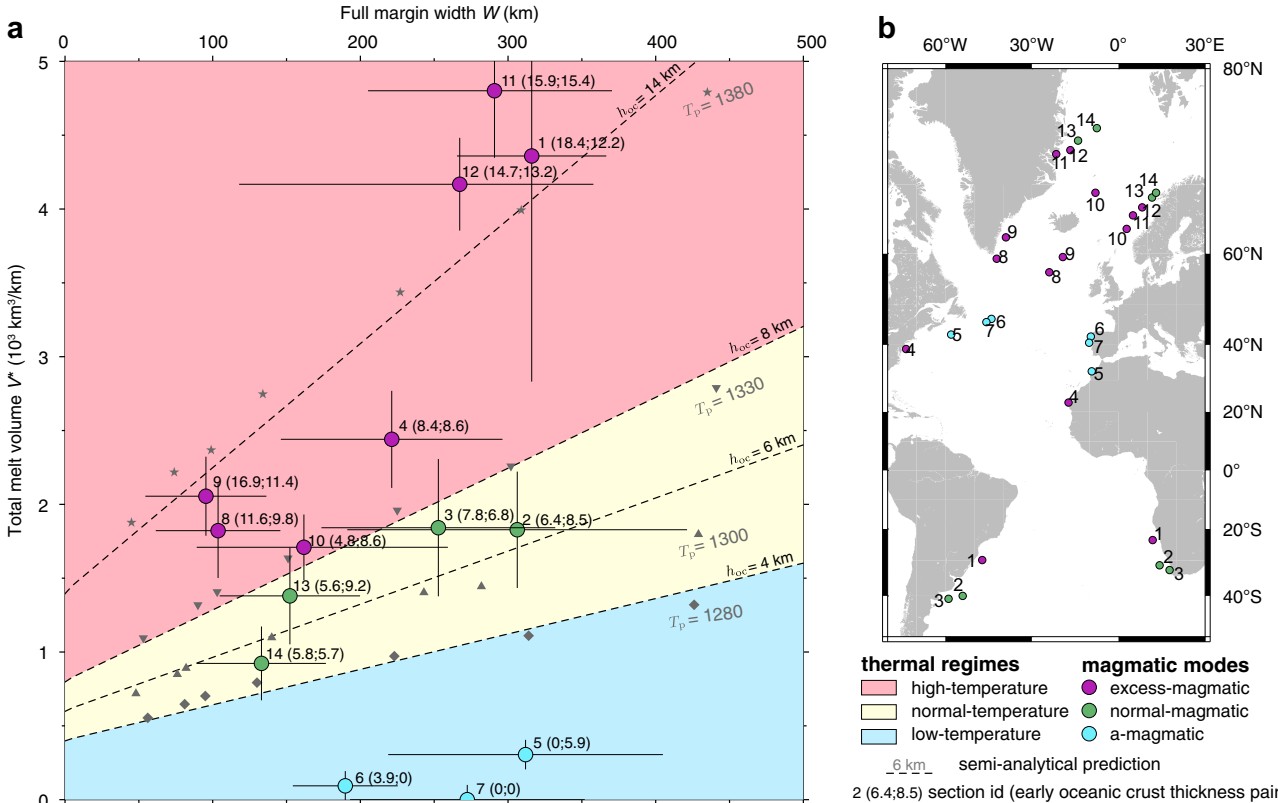

**Fig. 6 Comparison of observed versus predicted melt volume and margin width. a** Melt volume–margin width regimes. Grey symbols: model results at various $T_p$ labelled in degree Celsius; dashed lines: semi-analytical prediction for different oceanic crust thickness in km. Filled regions show thermal regimes for low-temperature (light blue), normal-temperature (pale yellow) and high-temperature (pink). Colour dots: observation data (Table 1) for a-magmatic (blue), normal-magmatic (green) and excess-magmatic (magenta) modes, with their uncertainties explained in Supplementary Table 2. **b** Locations of conjugate rifted margins with colours shown in **a**. See Supplementary Figs. 5–7 and Supplementary Table 2 for details.

Greenland–Hatton Bank section, White et al[8]. estimated excess temperatures of ~150 °C at Hatton Bank, with no requirement for significant active small-scale mantle convection. Brown and Lesher[61] suggest that mantle temperature for the Hatton Bank is elevated by 125 °C in combination with significant active mantle upwelling. Holbrook et al.[22] suggest that the thermal anomaly at breakup in the North Atlantic was ~100–125 °C in combination with moderate active upwelling. Numerical models[10,17] show that a 50-km-thick hot horizontal layer with excess temperature of 200 °C may lead to a magmatic pulse resulting in an igneous crustal thickness distribution comparable to observations at along the SE Greenland margin. Our models, with depth-dependent extension, provide an alternative scenario that not only predicts the magmatic pulse at breakup but also provides a mechanism for previously inferred high rates of active upwelling at volcanic rifted margins[22,61].

The semi-analytic scaling law and the numerical models presented here provide a new framework for understanding the variation of magmatic accretion during volcanic rifted margin formation. We show that while narrow margins with normal potential temperature mantle are expected to lead to a sharp transition from thinned continental crust to normal thickness oceanic crust (Fig. 7a), depth-dependent extension with preferential removal of the mantle lithosphere results in early melt addition in wide margins without requiring anomalously high mantle temperature (e.g. Fig. 7b). This provides an explanation for large volumes of magmatic accretion such as observed along some volcanic rifted margins[6,19], where plume activity cannot be easily demonstrated. The combined effect of depth-dependent extension and a small mantle temperature anomaly explain the

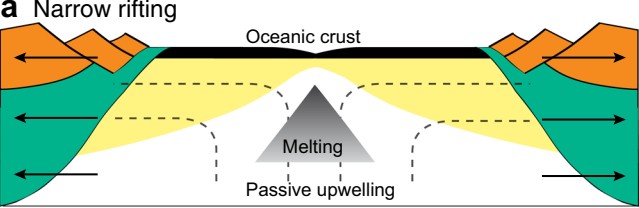

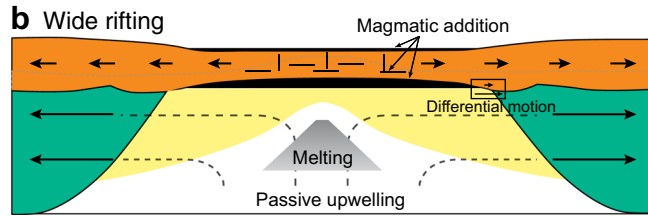

**Fig. 7 Cartoon comparing magmatic outputs of narrow and wide rifting at normal mantle temperature. a** Narrow rifting with simultaneous rupture of crust (orange) and mantle lithosphere (green), followed by accretion of normal oceanic crust (black). **b** Wide rifting with distributed deformation in the crust (orange) and narrow rupture of the mantle lithosphere. Note the differential motion between the crust and mantle lithosphere in wide rifting. Preferential removal of the mantle lithosphere leads to accumulation of magmatic addition (black) to the extending continental crust above. Note both narrow and wide rift scenarios have the same (normal) mantle potential temperature. Shown are melt window (grey region), new lithosphere (yellow), directions of extension (arrows) and flow path of passive mantle upwelling (dashed lines).

variation of magmatism along North, Central and South Atlantic rifted margins. We note that in cases where plume involvement is required to explain the observed magmatic volume, a very moderate mantle temperature anomaly in the order of 50–80 °C is sufficient, significantly smaller than previously suggested[5,8].

## Methods

**Thermo-mechanical model.** The forward numerical models of rifted margin formation are conducted using finite-element code SOPALE[62] to model upper mantle scale geodynamic processes[14,24]. The code solves thermo-mechanically coupled viscous-plastic creeping flows and uses Arbitrary Lagrangian–Eulerian approach to track material properties. A particle-in-cell method is applied to resolve advection of material phases as well as track material properties such as accumulated strain. Re-meshing is applied at each time step to avoid large grid distortion and to track the free surface. Laboratory-based power-law creeping flow laws are used for viscous deformation, with effective viscosity specified by:

$$\eta = f A^{-\frac{1}{n}} \left( \dot{E}'_2 \right)^{\frac{1-n}{2n}} \exp\left( \frac{Q+PV}{nRT} \right) \quad (1)$$

where $n$, $A$, $Q$ and $V$ are laboratory-derived constants (see Supplementary Table 1), $P$ pressure, $T$ absolute temperature, $R$ the universal gas constant, $\dot{E}'_2 = \frac{1}{2}\dot{\varepsilon}'_{ij}\dot{\varepsilon}'_{ij}$ is the second invariant of the deviatoric strain rate and $f$ is a viscosity-scaling factor that is used to generate stronger or weaker materials[14]. Plasticity is implemented with the Drucker–Prager yield criterion, which is activated when the second invariant of the deviatoric stress ($J'_2 = \frac{1}{2}\sigma'_{ij}\sigma'_{ij}$) exceeds the yield stress

$$\sigma_y = \left( J'_2 \right)^{1/2} = C\cos\varphi_{\text{eff}} + P\sin\varphi_{\text{eff}} \quad (2)$$

where $\varphi_{\text{eff}}$ is the effective internal frictional angle and $C$ is cohesion. $\sin\varphi_{\text{eff}} = \left( P - P_f \right)\sin\varphi$, where $P_f$ is the pore fluid pressure, $\varphi$ is internal frictional angle at dry condition. $\varphi_{\text{eff}} \approx 15°$ corresponds to hydrostatic pore pressure. Strain weakening is applied by linearly decreasing the effective frictional angle from 15° to 2° for accumulated visco-plastic strain ranging from 0.5 to 1.5.

**Rheological model setup.** The initial model (Supplementary Fig. 1) has laterally homogeneous layers of crust (35 km), mantle lithosphere (90 km) and sub-lithospheric mantle (475 km) from top to bottom. The crust is divided into upper crust (25 km) and lower crust (10 km) for visualization purpose, both of which have the same properties. A weak seed is imposed to localize deformation in the model centre. The parameters used here are listed in Supplementary Table 1. Viscous creep laws for the crust and mantle are Wet Quartz[34] and Wet Olivine[63], respectively. Crustal strength is varied using the crustal viscosity-scaling factor $f_c$. The crustal viscosity-scaling factors for models I and II are $f_c = 30$ and $f_c = 0.02$, respectively. The model top is a free surface. The sides are free slip, and the base is a horizontal free slip boundary. Horizontal extension velocities of $\pm V_{\text{ext}}/2$ are applied to at side boundaries in the lithosphere and the corresponding exit flux is balanced by a velocity inflow in the sub-lithospheric mantle, $V_b$ (Supplementary Fig. 1).

**Thermal model setup.** The initial temperature field, which is configured analytically, is laterally uniform, and consists of three segments delimited at Moho ($z_m$) and base lithosphere ($z_l$). The sub-lithospheric mantle follows an adiabatic geothermal gradient, 0.4 °C/km, with given potential temperature, $T = T_p + \frac{dT_a}{dz}z$. For the reference model with $T_p = 1300$ °C, this leads to base lithosphere temperature of $T_l = 1350$ °C at depth 125 km. The initial geotherm in the mantle lithosphere is linear between Moho temperature, $T_m = 550$ °C, which is configured to be the same for all models, and base lithosphere temperature $T_l$. The initial temperature in the crust increases with depth from the surface, $T_0 = 0$ °C, to the base of the crust ($T_m = 550$ °C), and follows a stable continental geotherm, $T = -\frac{A_r}{2k}\left( z - z_m \right)z + \frac{T_m}{z_m}z$, for uniform crustal heat production $A_r = 0.88$ μW/m³, which results in a basal heat flux, $q_m = 20$ mW/m² that matches the heat flux in the mantle lithosphere (i.e. steady state in the lithosphere). For models with a higher or lower potential temperature, and therefore different base lithosphere temperature $T_b$, heat production in the crust is adjusted to match the heat flux in the mantle lithosphere. Thermal boundary conditions are specified surface temperature for the top (0 °C) and bottom (1540 °C for the reference model) boundaries, and insulated side boundaries. The value of the bottom boundary temperature is adjusted according to potential temperature. Latent heat of melting and adiabatic heating/cooling is taken into account. Thermal diffusivity, $\kappa = k/\rho c_p = 10^{-6}$ m²/s.

**Melt parameterization model.** We use a parameterized melt prediction model[24], based on refs. [64,65]. Incremental melt fraction in each time step is calculated as:

$$d\phi_m = \frac{T - T_s}{L + \frac{\partial T_s}{\partial \phi_m}} \quad (3)$$

where $T$ is mantle temperature, $T_s$ is solidus temperature and $L = \frac{T\Delta S}{c_p}$ is latent heat, $c_p$ the heat capacity and $\Delta S$ the change of entropy on melting (Supplementary

Table 1). The solidus temperature is parameterized as a function of depth ($z$) and compositional depletion ($X$) (ref. [64])

$$T_s = T_{s0} + \frac{\partial T_s}{\partial z}z + \frac{\partial T_s}{\partial X}(X - 1) \quad (4)$$

where $T_{s0}$ is the solidus temperature at the surface. The compositional depletion represents the concentration of perfectly compatible elements in the solid phase and evolves with melting as

$$X\left( 1 - \phi_m \right) = 1 \quad (5)$$

Damp melting is included and is linearly parameterized[24] to be 0 on the wet solidus ($T_{sw} = T_s - 200$) and $\phi_{\text{lim}} = 0.02$ on the dry solidus ($T_s$). Although damp melting occurs at greater depth than dry melting, melt production is dominated by dry melting because water as an incompatible component is rapidly exhausted when melt fraction reaches $\phi_{\text{lim}}$. We track total predicted melt thickness at the surface. When the melt fraction exceeds the melt retention threshold of $\phi_{\text{ret}} = 0.01$, the extra melt is added to equivalent melt thickness that is tracked using a separate set of Lagrangian collection particles moving at surface velocity[24]. The melt fraction retained in the host rock ($\phi_m < \phi_{\text{ret}}$) is assumed to lead to a density feedback ($\Delta\rho_m = -(\rho_0 - \rho_m)\phi_m$, where $\rho_0$ and $\rho_m$ are mantle reference density and melt density, respectively) and a viscosity feedback ($\Delta\chi_m = \exp(-a\phi_m)$, where $a$ is an empirical constant[65]). Mantle melting also leads to a density change owing to depletion ($\Delta\rho_X = -\frac{\rho_0 - \rho_{X\text{ref}}}{X\text{ref} - 1}(1 - X)$, where $\rho_{X\text{ref}}$ is the density of residual mantle at reference depletion $X_{\text{ref}}$) and a viscosity change owing to dehydration during damp melting ($\Delta\chi_{\text{OH}} = \frac{5-1}{0.02}\phi_m + 1$, for $\phi_m < \phi_{\text{lim}}$, where $\phi_{\text{lim}} = 0.02$ is the maximum melt fraction for damp melting).

**Semi-analytical scaling law.** Analysing the underlying physics of depth-dependent wide rifting allows us to establish the linear correlation. If all the melt generated in the melting regime forms oceanic crust immediately, then the total melt produced during each increment of spreading equals the thickness of oceanic crust[2]. In other words, oceanic crustal thickness ($h_{\text{oc}}$) describes the quantity of melt produced per unit distance of spreading. In the case of wide rifting, before final breakup, we can define the effective melt thickness, $h_{\text{eff}}$, as the quantity of melt produced per unit distance of extension. $h_{\text{eff}}$ is smaller than $h_{\text{oc}}$ because upwelled mantle experiences lower degree of melting during continental rifting than during mid oceanic ridge spreading. The degree of melting is controlled by the height of upwelling mantle at temperatures above solidus[2] (Supplementary Fig. 4c), which is dominated by the thickness of the conductive thermal lid above[66]. In our models, most decompression melting is produced during dry melting, which occurs in a triangle domain (melt window) with its base at a depth of ~60 km for normal mantle potential temperature (Fig. 2). The height of melt window is smaller during rifting ($d_r$) than during spreading ($d_s$) (Supplementary Fig. 4a, b). Although the effective melt thickness $h_{\text{eff}}$ can not be directly constrained, the ratio between $h_{\text{eff}}$ during rifting and $h_{\text{oc}}$ during spreading can be calibrated by comparing the heights of their melt windows as $h_{\text{eff}}/h_{\text{oc}} \cong d_r/d_s \cong 0.6$ for reference potential mantle temperature $T_p = 1300$ °C (Supplementary Fig. 4). Consequently, total melt volume, including the contribution from the 100-km initial spreading section, may be expressed as

$$V^* = 0.6h_{\text{oc}}W + 100h_{\text{oc}} \quad (6)$$

Assuming constant melt productivity during rifting and spreading, respectively, Supplementary Fig. 4d conceptually illustrates how the total melt volume is dependent on margin width and mantle potential temperature.

## Data availability

All model parameters are available in Supplementary Table 1. The data for this paper, including model data for the plots and plotting scripts, can be accessed from Pangaea Data Archiving and Publication (https://doi.org/10.1594/PANGAEA.905111).

## Code availability

The source code to calculate parameterized melt fraction can be accessed from Pangaea Data Archiving and Publication (https://doi.org/10.1594/PANGAEA.905111).

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

## Acknowledgements

We are grateful to reviewers Jenny Collier and Tobias Keller for their very constructive comments. We acknowledge support of the Department of Earth Science, University of Bergen, Norway. This study is funded through the COLORS project by Total. Computational resources have been allocated thanks to Uninett Sigma2 for project NN4704K.

## Author contributions

G.L. contributed the numerical models and data collection for volcanic margins. R.S.H. contributed ideas on rifted margin styles. Both authors contributed to developing the concepts and to writing the manuscript.

## Competing interests

The authors declare no competing interests.
