## [Peer Review File · Nature Communications]

REVIEWER COMMENTS

Reviewer #1 (Remarks to the Author):

This manuscript presents new numerical modelling work to predict melt volumes during continental breakup. This is a topic of broad interest, both for the study of rifted margins (and their economic potential) themselves but also as they provide constraints on the properties of the mantle which are otherwise hard to obtain. The topic also has significance for Earth history in terms of the environmental consequences of such significant pulses of volcanism.

The manuscript is well written and illustrated and will make a nice contribution to Nature Comms. My main comments are as follows:

The manuscript seems to take the assumption that narrow rifts indicate uniform extension whereas wide rifts indicate depth-dependant extension (line 63). It is not totally clear where this assumption comes from, and possibly the general reader will become confused by the relationship with the senior author's previous work on non-volcanic margins? I understand this comes out in the results (line 98) – but the argument seems a bit circular at present. The issue of asymmetry (in which a wide-rift can be observed on one side and a narrow-rift on the conjugate e.g. SIGMA-3 – iSIMM Hatton; North Greenland AWI - Lofoten) is very much glossed over. Whilst it is mentioned on line 141 it is not commented in, for example, table 1 or ED table 2 where the observations are compiled.

One observation that has not been discussed is the thinning of the initial oceanic crust over time in the volcanic margin case. This is seen in the real example (Walvis) presented it is ignored in the cartoon (Mode 2) in Fig 1, it is also in Table S2 (but not commented on?). On line 84 it states that this thinning happens within 30 km. In the South Atlantic I would say it was more like 100 km (Taposeea et al., 2016). This reduction in productivity post-breakup is the reason why in the modelling work led by John Armitage we use an exhaustible hot layer below the mantle lithosphere (to simulate an impacted plume head). This distinction should be made as the temperature estimates in the work presented here come from a bulk value for the entire sub-lithospheric mantle in the models (e.g. for the South American margins on line 193 the cited 36 reference 750C excess asthenospheric temperature is confined to a 100 km thick layer).

With the caveats given above – the results relating margin width to melt productivity are none-the-less interesting. I feel the manuscript lacks a brief comparison of the results with other measures of mantle potential temperature during breakup – such as lava geochemistry.

Specific points:

Title – not very gripping – perhaps give a better summary of the key result. Similarly on line 23 of the abstract give the proposed reduction compared to earlier studies (instead of "much smaller"). For example, in Armitage & Collier 2017 Petroleum geosciences where we summarise the results from 4 locations analysed with a non-depth dependant extension, we obtained a range of 100-200C, so 50-80C is arguably "half the value previously suggested".

Line 29. In Armitage et al., 2010 we make the case that the pre-thinning in the North Atlantic ("rift history") was partly responsible for the extreme excess magmatism here.

Line 78 and ED Fig 2. There appears to be a positive gradient of crustal thickness vrs spreading rate between the 50 and 100 mm/yr points. Based on observations it is usually assumed to be flat at a given mantle potential temperature. The authors should comment on this (and maybe calculate some intermediate points). Also the Bown & White 1994 data points are a bit old – Christensen 2019 JGR or Grevemeyer 2018 Geosphere exclude old ESP-type measurements which are problematic.

Line 219. Are there any geological reasons why the lower crust of the South Atlantic province would be weaker than the North Atlantic province (to give the required depth-dependant extension difference)?

Fig 2 Define f_c here (don't wait for Fig 3). Is there an inconsistency with the shading shown in ED Fig 1b – where model 1 (strong lower crust) is all orange i.e. white layer represents weak lower crust. Does the thickness of the weak layer change? An interesting aspect of the models shown in Fig 2 is that for the model 2 case, the weak lower crust becomes “swept up” into the upper crust. Do the authors think this is real or just an artefact of the model? Is it random? It is interesting as it could account for internal crustal reflectivity seen on some very-long offset commercial lines.

Fig 3 Isn't extension rate more important than total extension (for conductive cooling)? Are all the models run at the same extension rate?

Fig 7 and lines 214-216 Are the mantle temperatures identical (i.e. is normal the same as not anomalous!) – be a bit more explicit.

All references should be checked as there are a lot of proper noun capitals missing.

Jenny Collier (London, 4/1/21)

Reviewer #2 (Remarks to the Author):

This manuscript presents thermo-mechanical models of continental rifting to oceanic spreading using a calibrated parameterisation of mantle melting to predict the equivalent thickness of igneous crust produced by extracted melt. With this model, the authors address the question what factors control the observed regimes of excess magmatic, normal magmatic, and a-magmatic rift margins. Their analysis shows that, as expected, mantle potential temperature controls the total amount of magmatism and hence the oceanic crustal thickness once the rifting has transitioned to mature oceanic spreading. More interestingly, the models show that decoupling between rapid rifting of strong continental lithosphere and slow rifting of a weak crust leads to more magmatism concentrated within the extended margin. The authors show that this trend can be reasonably well captured by a semi-analytical fitting function predicting a linear relationship between margin width and total magmatic volume produced during rifting. Their predictive model is demonstrated to be useful for classifying and explaining observed differences in levels of magmatism for various natural rift systems.

Overall, the manuscript is of a high quality in terms of research method, analysis of results, and discussion in the context of observations. The conclusions shed new light on the long-standing discussion around factors

controlling abundance of magmatic products on rifted margins. Although there is room for improvement surrounding the clarity of some of the analysis as presented in the main text, and perhaps some need to better communicate the simplified nature and hence limitations of the modelling approach, I believe this research can be published after minor revisions. The research is timely and at the leading edge of the respective field. It is of broad interest to the geoscience community and would therefore be a good fit for the journal. In the following I will detail a number of issues and corrections I would recommend the authors address.

Minor Issues

Line 47 It would perhaps be useful to the generalist reader to briefly mention some examples of magma-rich margins where plume influence has been ruled out.

Line 54 The authors could perhaps describe a bit more what uniform lithospheric thinning would involve, e.g. does uniform thinning require stretching by pure-shear creep, or a set of lithospheric-scale symmetrical normal faults, etc.?

Lines 56-57 This passage makes use of what sounds like jargon to myself and possibly other non-experts in rift modelling. Terms like coupled or decoupled deformation and thinning factor are not self-explanatory and require careful definition or else should be replaced by more descriptive language.

Line 60 I stumbled over the expression these end member styles here because the authors introduce two sets of end-member regimes in text so far. One is the observed end-members of excess-, normal-, and a-magmatic rifts, and the second are narrow, uniform, coupled versus wide, depth-dependent, decoupled rifts. I think there is room to improve clarity and more clearly point out that the former set of three is tied to observations, whereas the second two follow from model interpretations. If I understood correctly, the hypothesis set up in this work is that the two model end-members explain the variability between the three observed magmatic regimes. I think this can still be further clarified in this passage.

Line 75 What exactly do the authors mean by crustal strength? Is it viscosity, elastic modulus, or yield stress? Or a combination of these?

Line 75 Melt prediction model: I am sure the authors are aware that prediction is a somewhat loaded term. If I understand correctly, this is a parameterisation calculating the amount of decompression melting produced by upper mantle flow. I would therefore recommend using the term 'melt parameterisation model' to better convey the nature of this treatment. It would also serve the readers' understanding to briefly but clearly state the simplifications and consequent limitations of this approach. Most importantly, this treatment does not allow for magmatism to interact with tectonic deformation and cannot distinguish between eruptive and intrusive magma emplacement.

Line 77 Melt thickness: this term is not self-evident to the non-expert reader and needs to be clearly defined when first used.

Line 81 Strong crust: what is the controlling parameter for crustal strength? the figures mention f_c , which I think should be briefly introduced here even if the details are later stated in the method section.

Line 83 Efficient rupture: what does efficient mean in this context? Can this be put in more descriptive language?

Line 87 Upper and lower lithosphere: it is not entirely clear which layers are referred to here, is it the crust and mantle lithosphere?

Lines 89-90 Early mantle lithosphere rupture: I did not quite understand what the authors mean here. Fig. 2 a&b show snapshots at equal model time, and the mantle lithosphere appears separated by about the same distance. It would therefore look like the lithosphere ruptured at around the same timing? What then is the meaning of early?

Line 92 The authors describe the higher peak melt thickness produced beneath the distal margin upon first rupture of the mantle lithosphere. It remains somewhat unclear why there should be higher melt production when the lithosphere fails beneath a stretched crust. Higher melt production would indicate higher rates of upward flow in the asthenosphere. How exactly is that flow controlled by the coupled versus decoupled rifting styles? A little further along on line 95, the authors say the increased magma production is explained by depth-dependent extension and the larger extent of lateral advection of the continental mantle lithosphere. However, lateral advection does not produce melt, but vertical flow does. How are they connected here? Why does the case with weak crust induce three times as much mantle decompression flow? There is also conspicuous asymmetry in melt distribution not present in model I. I recommend the authors try to clarify the process-based causality between margin width and increased magma production.

Line 101 Linear correlation between margin width and total magmatic volume: this result is difficult to follow because up to here the metrics for margin width and total magmatic volume have not been clearly introduced. A brief definition should be stated in the main text when the terms are first used even if details of the processing are given in the Method section below. Also, it should be stated that the relationship is approximated by a linear trend (I presume for simplicity), despite the metrics in Fig. 4 showing a tendency to a more square-root or power-law-like relationship.

Lines 109-110 Melt window, thermal lid: this passage reads too much like jargon with terms that are not clearly defined or self-evident. The authors should carefully consider that this journal addresses a general audience.

Line 112 A thicker thermal lid ... results in lower melt production: I stumbled over this sentence because up to now the results highlighted that wider margins produce more melt, but here it is stated that wider margins result in a thicker lid and lower melt production. How is this resolved?

Lines 113-118 This passage is perhaps the weakest point of the manuscript. I did not understand much of how the authors arrived at their semi-analytical model, or how the different metrics referred to here are extracted from the model results. Since this analysis is the connecting point between models and observations it is crucial to the success of this study to render this as clear as possible. As it stands, a number of terms and symbols are not clearly defined in the text (melt thickness, extent of melting, d_r and d_s). The relationship between total melt volume, margin width and mature oceanic crustal thickness (related to mantle temperature, right?) is clearly not a complicated one, indeed it would appear to be a simple linear fit to the data points in Fig. 4. However, I still don't follow what exactly this metric represents and how the authors

arrived here. (After writing this comment I read on to the end of the manuscript, where these matters are quite helpfully visualised in the Extended Data Figures. I would strongly recommend to rewrite this paragraph for clarity, to consider whether some of the conceptual visualisation of this analysis could be added to main text figures, or to at least clearly refer to the Extended Data Figures if that is where the related information remains.)

Line 122 Slope of the linear relationship, $heff$: Up to this point it had not become clear to me that $heff$ is the slope of that linear relationship. This should be further clarified.

Line 122 Predicts and confirms: this semi-analytical scaling law cannot predict and confirm the relationship between margin width and magma production at the same time. Normally, a model would make a prediction that may be confirmed by observations or by an independent experiment/model. Here it seems that the authors derive a simplified relationship that fits their model results. If that is correct it would not qualify as a confirmation, only as a derivative predictive metric.

Line 127 It remains unclear how heq and hoc are related.

Line 151ff Total melt volume: it appears that total melt volume has units of volume per length, although that is not clearly stated anywhere. I recommend to adapt the terminology to convey that this is not simply a total volume but a total volume produced per margin width. In this passage the meaning of the metric V^* above becomes a little clearer. I think it is vital to clarify exactly what the model metric above represents and why it is meaningful for comparison with natural data when it is first introduced above.

Line 158ff Margin width: if this is the same metric as used to evaluate margin width in model results then this definition should be introduced further above where the metric is first introduced. As stated above, the authors might consider adding some of the conceptual visualisation of these metrics to a main text figure or else should clearly refer to the Extended Data Figure with that information here.

Lines 183ff Text structure: the first paragraph of the Discussion section reports the results of the data analysis on natural rifted margin and would therefore sit better in a Results section. Perhaps the authors could reconsider their section structure and form sections for Model Results, Semi-analytical Scaling Law, and Natural Rift Classification, before ending the main text on a brief Discussion.

Line 187 4-8 km oceanic crust: it appears that the authors here use oceanic crustal thickness to discriminate between magmatic margin regimes. This is also what Fig. 6 confirms. On the other hand, oceanic crustal thickness is taken as proxy for mantle potential temperature. It would therefore appear that excess-, normal-, and a-magmatic regimes are determined by mantle potential temperature, not margin width. This seems to run counter to the overall conclusion of the article that margin width is the important controlling parameter, and that mantle potential temperature has secondary effects. It is possible that I have misunderstood part of the argument here but it would certainly be worth to clarify the relative roles of T_p and W in classifying ridge regimes both in the model and in natural data.

Fig. 5 This Figure is very helpful for understanding how analytical metrics are extracted from observations. A similar figure would be helpful for how metrics are extracted from model results as well.

Line 482 Accumulated strain: does this refer to total visco-plastic strain or only plastic failure strain?

Line 484 It is not made clear in the text what experimental creep parameters are used for the crustal and mantle layers. In the model setup figure only wet quartz and wet olivine are listed. Does that mean that upper and lower crust have the same rheology? If so, what else distinguishes these layers?

Line 512 Is latent heat consumption also included in the energy equation used to solve for T? If so then it should be noted in the paragraph on the thermal model.

Extended Data Figures are very helpful and clarify a number of questions that came up while reading the main text. I encourage the authors to consider if some of that information (particularly the visualisation of analytical metrics used to derive the semi-analytical scaling law) could be transferred to the main text. Either way, these helpful figures should be more explicitly referred to in the main text.

Best Regards,

Tobias Keller (Tobias.Keller@glasgow.ac.uk)

We appreciate the two reviewers for their positive and very constructive comments. In this file we reply to all the comments point by point as listed below. Original comments are in black and our replies are in blue.

REVIEWER COMMENTS

Reviewer #1 (Remarks to the Author):

This manuscript presents new numerical modelling work to predict melt volumes during continental breakup. This is a topic of broad interest, both for the study of rifted margins (and their economic potential) themselves but also as they provide constraints on the properties of the mantle which are otherwise hard to obtain. The topic also has significance for Earth history in terms of the environmental consequences of such significant pulses of volcanism.

The manuscript is well written and illustrated and will make a nice contribution to Nature Comms. My main comments are as follows:

The manuscript seems to take the assumption that narrow rifts indicate uniform extension whereas wide rifts indicate depth-dependant extension (line 63). It is not totally clear where this assumption comes from, and possibly the general reader will become confused by the relationship with the senior author's previous work on non-volcanic margins? I understand this comes out in the results (line 98) – but the argument seems a bit circular at present. The issue of asymmetry (in which a wide-rift can be observed on one side and a narrow-rift on the conjugate e.g. SIGMA-3 – iSIMM Hatton; North Greenland AWI - Lofoten) is very much glossed over. Whilst it is mentioned on line 141 it is not commented in, for example, table 1 or ED table 2 where the observations are compiled.

We understand the confusion. There is no a-priori assumption that narrow rifts exhibit uniform extension and indeed both narrow and wide margins are characterised by depth dependent extension. The models presented here build upon earlier work by Huisman (2011, 2014). In these studies, Type I narrow margins and Type II wide margins are referred to as end-member tectonic styles that are both depth-dependent. The key difference is that Type II margins have preferential removal of mantle lithosphere whereas Type I margins have preferential removal of the crust. A strong crust is responsible for early rupture and preferential removal of the crust at Type I narrow margins. We have revised the manuscript to avoid using "uniform" when referring to narrow margins.

Final asymmetric rupture of wide rifted margins is indeed commonly observed both in the North and South Atlantic, and also reproduced in numerical models (Huisman & Beaumont, 2003; Brune et al., 2014). It is also observed in our current work, as shown in Extended Data Figure 3 for the model with $f_c = 0.1$. Asymmetry is promoted

by the degree of strain weakening (Huisman & Beaumont, 2003): strain softening breaks symmetry and promotes asymmetry through the positive feedback between softening and strain accumulation. What is important is that asymmetry only alters the distribution of magmatic crust on margins, but does not change the total width of the conjugate margins and the total melt volume produced during the whole rifting process. We therefore estimate and compare the total melt volume from both conjugate margins in order to minimize the influence of asymmetry. The wide rifting models predict the same narrow rupture of the mantle lithosphere during crustal thinning, whether the margin system evolves as symmetric or asymmetric. As a result melt enhancement is independent of the (a) symmetry of the rift system, which is why we initially did not emphasise this. However, we understand that this can lead to confusion and have therefore included a comment on this in the revised manuscript.

One observation that has not been discussed is the thinning of the initial oceanic crust over time in the volcanic margin case. This is seen in the real example (Walvis) presented it is ignored in the cartoon (Mode 2) in Fig 1, it is also in Table S2 (but not commented on?). On line 84 it states that this thinning happens within 30 km. In the South Atlantic I would say it was more like 100 km (Taposeea et al., 2016). This reduction in productivity post-breakup is the reason why in the modelling work led by John Armitage we use an exhaustible hot layer below the mantle lithosphere (to simulate an impacted plume head). This distinction should be made as the temperature estimates in the work presented here come from a bulk value for the entire sub-lithospheric mantle in the models (e.g. for the South American margins on line 193 the cited 36 reference 750C (correction: we think this should be 150C here) excess asthenospheric temperature is confined to a 100 km thick layer).

We are well aware that oceanic crustal thickness decreases to normal values after breakup at volcanic rifted margins. Keen and Boutilier (2000) and Nielsen and Hopper (2004) have explored the role of a thin hot plume layer at the base of the lithosphere. Armitage et al. (2010) using the same approach focused among other things on the compositional effects and rift history. These models with a finite reservoir of hot mantle material may explain the decrease from excess to normal thickness igneous crust. For the purpose of our study, a hot plume layer and a uniform hot mantle have the same effect in enhancing melt volume during breakup at volcanic margins. We did not include the potential effects of a hot plume layer in our modelling approach for simplicity reasons. We have, however, added comments on these earlier models in the revised manuscript.

With the caveats given above – the results relating margin width to melt productivity are none-the-less interesting. I feel the manuscript lacks a brief comparison of the results with other measures of mantle potential temperature during breakup – such as lava geochemistry.

We agree that it is useful to include a brief discussion of the estimates of T_p from other studies. We have added these to the discussion in the revised manuscript. We note that estimates of mantle potential temperature based on geochemistry show large variations and are not easy to calibrate.

Specific points:

Title – not very gripping – perhaps give a better summary of the key result. Similarly on line 23 of the abstract give the proposed reduction compared to earlier studies (instead of “much smaller”). For example, in Armitage & Collier 2017 *Petroleum geosciences* where we summarise the results from 4 locations analysed with a non-depth dependant extension, we obtained a range of 100-200C, so 50-80C is arguably “half the value previously suggested”.

We have revised the title, which we believe gives a better summary of the results.

We have included a discussion of these earlier studies, and suggest that “our estimated T_p is significantly smaller than previously suggested.”

Line 29. In Armitage et al., 2010 we make the case that the pre-thinning in the North Atlantic (“rift history”) was partly responsible for the extreme excess magmatism here.

We refer to this hypothesis and to Armitage et al., 2010 in the revised manuscript.

Line 78 and ED Fig 2. There appears to be a positive gradient of crustal thickness vrs spreading rate between the 50 and 100 mm/yr points. Based on observations it is usually assumed to be flat at a given mantle potential temperature. The authors should comment on this (and maybe calculate some intermediate points). Also the Bown & White 1994 data points are a bit old – Christensen 2019 *JGR* or Grevemeyer 2018 *Geosphere* exclude old ESP-type measurements which are problematic.

We believe the slight increase in igneous crustal thickness with spreading rate is an expected behaviour in numerical models. Lower spreading rates result in lower upwelling velocity and thicker thermal conductive lid, leading to decreased melt production rate and thinner igneous crust. This effect is particularly significant for slow spreading ridges, but takes also place for faster spreading ridges.

Similar increase in predicted igneous crust is also shown in earlier modelling studies (Armitage, 2008; Forsyth, 1992; Nielsen & Hopper, 2004). For example, in the work by Nielsen & Hopper (2004), there is also a slight increase of crustal thickness versus spreading rate (Fig. 1A). In the study of Armitage (2008, PhD Thesis), modelled igneous thickness also increases with faster spreading rate (Fig. 1B), albeit with a more significant positive gradient than in this study.

Figure 1. Earlier studies of comparison between predicted igneous thickness and observations. Shown examples are Nielsen and Hopper (Nielsen & Hopper, 2004) (A) and Armitage (2008)(B).

We are aware that the Bown & White (1994) data points are a bit old. Comparison with the newer data set of Grevemeyer et al. (2018) shows that our model results fit the observations reasonably well (Figure 2). We would, however, prefer to keep the comparison with the data originally compiled by Bown & White (1994).

Figure 2. Benchmark of predicted igneous thickness in this study with observation data by Grevemeyer et al. (2018).

We believe that our model approach is robust and captures the melting behaviour during continental rifting and seafloor spreading. There may be other processes that break the positive gradient in natural systems, which are not included in our models.

Line 219. Are there any geological reasons why the lower crust of the South Atlantic province would be weaker than the North Atlantic province (to give the required depth-dependant extension difference)?

Both the North Atlantic and South Atlantic margins are very wide. Geodynamic models suggest that such wide margins require weak crustal rheology. The main difference between the North and South Atlantic is that the North Atlantic margins have experienced poly-phase rifting over a very long time period, whereas rifting leading to breakup in the South Atlantic occurred within a timeframe of about 25 Myr's. The final breakup-related margin width in the North Atlantic, the one we measured, is narrower than South Atlantic margins presumably because the crust at the time of the final rift phase was thinned and therefore coupled to the underlying mantle lithosphere. As also suggested by England (1983) earlier rifting may lead to strengthening of the lithosphere due to cooling and result in migration of extension, which may explain why the last rifting phase in the North Atlantic is narrow.

Fig 2 Define f_c here (don't wait for Fig 3). Is there an inconsistency with the shading shown in ED Fig 1b – where model 1 (strong lower crust) is all orange i.e. white layer represents weak lower crust. Does the thickness of the weak layer change? An interesting aspect of the models shown in Fig 2 is that for the model 2 case, the weak lower crust becomes “swept up” into the upper crust. Do the authors think this is real or just an artefact of the model? Is it random? It is interesting as it could account for internal crustal reflectivity seen on some very-long offset commercial lines.

We have revised the main text to define f_c before introducing Fig. 2.

The upper crust and lower crust have the same properties. We used two materials (with different colours) in the model only for visualization purpose. We have revised the manuscript and explicitly mention that the upper crust and lower crust have the same properties.

The behaviour that the weak crust is “swept up” into the upper crust is as expected in weak and wide rifted margin cases. In Type-II models, the thin upper crust undergoes (localized) brittle deformation while the middle and lower crust deform in viscous manner (see the strength profile in Extended Data Figure 1). With on-going deformation, the ductile middle and lower crust may flow into the space created by brittle failure in the upper crust, leading to exhumation of mid to lower crust and formation of metamorphic core complexes. This behaviour has for instance been

observed on long offset lines for the Gabon margin (e.g., Clerc et al., 2018) and in earlier modelling work (e.g., Huisman & Beaumont, 2011, 2014; Theunissen & Huisman, 2019). Metamorphic core complexes provide also important geological evidence for weak crust in regions with wide rifting, such as for instance the Basin and Range (Jones et al., 1992), the central South Atlantic (e.g., Clerc et al., 2018), and the South China Sea (Deng et al., 2020).

Fig 3 Isn't extension rate more important than total extension (for conductive cooling)? Are all the models run at the same extension rate?

The extension rate is indeed important for the thickness of igneous crust for velocities lower ~ 1 cm/yr. To avoid the issue of conductive cooling the models are run at 1.5 cm/yr total extension/spreading rate. The models in Fig. 3 (and in Extended Data Figure 3) are all running at the same extension rate. Also, they all have the same potential temperature. The only difference between these models is the crustal scaling factor, and hence their margin width.

We show the models at the same total amount of extension as this allows comparing them at the same time. We have updated Fig. 3 and its figure caption and added the time and total extension in each subplot.

Fig 7 and lines 214-216 Are the mantle temperatures identical (i.e. is normal the same as not anomalous!) – be a bit more explicit.

The two models have the same 'normal' mantle potential temperature, so as to highlight the contribution of margin width. We revised the figure caption to clarify this.

All references should be checked as there are a lot of proper noun capitals missing.

We have fixed the references and added missing capital letters where needed.

Jenny Collier (London, 4/1/21)

Reviewer #2 (Remarks to the Author):

This manuscript presents thermo-mechanical models of continental rifting to oceanic spreading using a calibrated parameterisation of mantle melting to predict the equivalent thickness of igneous crust produced by extracted melt. With this model, the authors address the question what factors control the observed regimes of excess magmatic, normal magmatic, and a-magmatic rift margins. Their analysis shows that, as expected, mantle potential temperature controls the total amount of magmatism and hence the oceanic crustal thickness once the rifting has transitioned to mature oceanic spreading. More interestingly, the models show that decoupling between rapid rifting of strong continental lithosphere and slow rifting of a weak crust leads to more magmatism concentrated within the extended margin. The authors show that this trend can be reasonably well captured by a semi-analytical fitting function predicting a linear relationship between margin width and total magmatic volume produced during rifting. Their predictive model is demonstrated to be useful for classifying and explaining observed differences in levels of magmatism for various natural rift systems.

Overall, the manuscript is of a high quality in terms of research method, analysis of results, and discussion in the context of observations. The conclusions shed new light on the long-standing discussion around factors controlling abundance of magmatic products on rifted margins. Although there is room for improvement surrounding the clarity of some of the analysis as presented in the main text, and perhaps some need to better communicate the simplified nature and hence limitations of the modelling approach, I believe this research can be published after minor revisions. The research is timely and at the leading edge of the respective field. It is of broad interest to the geoscience community and would therefore be a good fit for the journal.

In the following I will detail a number of issues and corrections I would recommend the authors address.

Minor Issues

Line 47 It would perhaps be useful to the generalist reader to briefly mention some examples of magma-rich margins where plume influence has been ruled out.

We have revised the text and added examples of magma rich margins where the influence of a plume has been difficult to demonstrate or ruled out, e.g. the NW Australian margin (Hopper et al., 1992) and the US East Coast margin (Holbrook & Kelemen, 1993). The US East Coast margin example is included in our current study as line (4) Baltimore – Dakhla section.

Line 54 The authors could perhaps describe a bit more what uniform lithospheric

thinning would involve, e.g. does uniform thinning require stretching by pure-shear creep, or a set of lithospheric-scale symmetrical normal faults, etc.?

We have revised the manuscript and describe more clearly what is commonly assumed in earlier models.

Previous models of melt generation often treated the lithosphere as a whole, assuming that deformation in the lithosphere is nearly pure-shear. Our models here show that the tectonic style of rifting and margin formation (narrow versus wide margins) with non-uniform depth dependent extension has a significant influence on magmatic output.

Lines 56-57 This passage makes use of what sounds like jargon to myself and possibly other non-experts in rift modelling. Terms like coupled or decoupled deformation and thinning factor are not self-explanatory and require careful definition or else should be replaced by more descriptive language.

“Coupling” or “decoupling” here is referring to the interaction between the frictional-plastic upper crust and the frictional-plastic upper lithospheric mantle. “Coupled” means that the crust and upper mantle lithosphere deform together. “Decoupled” means that the crust and upper mantle lithosphere deform separately with a weak decoupling horizon (the low viscosity mid and lower crust) between them. The term thinning factor is now replaced by stretching factor, which should be more descriptive. The stretching factor is a classic measure used to quantify the degree of crust and mantle lithosphere thinning (or thickening) (McKenzie, 1978; Royden & Keen, 1980). The text has been updated to make the meaning of these terms more explicit.

Line 60 I stumbled over the expression these end member styles here because the authors introduce two sets of end-member regimes in text so far. One is the observed end-members of excess-, normal-, and a-magmatic rifts, and the second are narrow, uniform, coupled versus wide, depth-dependent, decoupled rifts. I think there is room to improve clarity and more clearly point out that the former set of three is tied to observations, whereas the second two follow from model interpretations. If I understood correctly, the hypothesis set up in this work is that the two model end-members explain the variability between the three observed magmatic regimes. I think this can still be further clarified in this passage.

Excess-, normal-, and a-magmatic margins are magmatic end-members. Narrow and wide margins are tectonic end-members. Both are tied to observations. For example, the Red Sea is a typical narrow rift, whereas the Central South Atlantic margins are typical ultra wide margins. The Iberia-Newfoundland system is the archetypical non-volcanic margin, whereas the North and South Atlantic margins are classic volcanic margins. Examples of these are listed in the manuscript and extended data.

To avoid confusion, we have revised the manuscript to refer to narrow versus wide margins as tectonic styles without stating “end-member” explicitly.

Line 75 What exactly do the authors mean by crustal strength? Is it viscosity, elastic modulus, or yield stress? Or a combination of these?

Crustal strength literally refers to the integrated frictional-plastic yield strength and viscous flow stress, e.g. the surface below the yield envelope, see Extended data Fig. 1 panel b for Model I (strong), and Model II (weak) rheologies with respectively $f_c = 30$ and 0.02 . In our models, we use the scaling factor f_c to change viscosity, which in turn alters the thickness of frictional/brittle layer and consequently changes crustal strength. f_c is a simple parameter to produce stronger or weaker crust, and is referred to as a proxy for crustal strength. We have revised the text to define f_c as the proxy for crustal strength.

Line 75 Melt prediction model: I am sure the authors are aware that prediction is a somewhat loaded term. If I understand correctly, this is a parameterisation calculating the amount of decompression melting produced by upper mantle flow. I would therefore recommend using the term ‘melt parameterisation model’ to better convey the nature of this treatment. It would also serve the readers’ understanding to briefly but clearly state the simplifications and consequent limitations of this approach. Most importantly, this treatment does not allow for magmatism to interact with tectonic deformation and cannot distinguish between eruptive and intrusive magma emplacement.

We agree with this and have revised “melt prediction model” to “melt parameterization model”.

Line 77 Melt thickness: this term is not self-evident to the non-expert reader and needs to be clearly defined when first used.

We have revised the manuscript and note that melt thickness is equivalent to “igneous crustal thickness” the first time it appears, which should be more self-explanatory.

Line 81 Strong crust: what is the controlling parameter for crustal strength? the figures mention f_c , which I think should be briefly introduced here even if the details are later stated in the method section.

See answer above about crustal strength. We agree and define both f_c and crustal strength earlier in the text.

Line 83 Efficient rupture: what does efficient mean in this context? Can this be put in

more descriptive language?

Efficient rupture means that the crust and mantle lithosphere localize deformation in a narrow manner leading to complete rupture after about 80-100 km of extension, in contrast to the wide rift model that requires > 350 km to complete rupture. We have revised the text to be more descriptive.

Line 87 Upper and lower lithosphere: it is not entirely clear which layers are referred to here, is it the crust and mantle lithosphere?

We have revised the text and refer to the crust and mantle lithosphere here.

Lines 89-90 Early mantle lithosphere rupture: I did not quite understand what the authors mean here. Fig. 2 a&b show snapshots at equal model time, and the mantle lithosphere appears separated by about the same distance. It would therefore look like the lithosphere ruptured at around the same timing? What then is the meaning of early?

The reviewer is correct. The mantle lithosphere ruptures at approximately the same time in both the narrow and wide rift models. It is the crust that has a different rupture time. "Early mantle lithosphere rupture" here is with reference to the rupture time of the crust. In wide rifting models, the strong mantle lithosphere ruptures earlier than the weak crust. We have revised the text to make this more explicit.

Line 92 The authors describe the higher peak melt thickness produced beneath the distal margin upon first rupture of the mantle lithosphere. It remains somewhat unclear why there should be higher melt production when the lithosphere fails beneath a stretched crust. Higher melt production would indicate higher rates of upward flow in the asthenosphere. How exactly is that flow controlled by the coupled versus decoupled rifting styles? A little further along on line 95, the authors say the increased magma production is explained by depth-dependent extension and the larger extent of lateral advection of the continental mantle lithosphere. However, lateral advection does not produce melt, but vertical flow does. How are they connected here? Why does the case with weak crust induce three times as much mantle decompression flow? There is also conspicuous asymmetry in melt distribution not present in model I. I recommend the authors try to clarify the process-based causality between margin width and increased magma production.

There is a misunderstanding here. The rate of (passive) mantle upwelling is the same for the narrow and the wide rift models and is controlled by the far field divergence rate. What is different is that in the narrow rift model the crust is moving at the same rate together with the mantle lithosphere below, whereas in the wide rift models the crust is moving at a lower rate. One can understand this by comparing

the horizontal length scale of corner flow (typically ~ 80-100 km) with the horizontal length scale over which quasi-pure shear extension is accommodated in the crust (300-400 km in the wide rift models). As the extension in the crust is distributed over a much larger area the horizontal velocity at which the crust moves is significantly lower compared to the rate of mantle upwelling below. The crust will therefore collect more melt, as it stays longer above the melt window. If for instance the crust moves at a rate 3 times lower than the corner flow mantle upwelling below, about 3 times the reference igneous thickness is consequently accumulated in the crust above.

Regarding the question of asymmetric melt distribution, corner flow mantle upwelling is largely symmetric. However, the deformation of the crust is not fully symmetric and pure shear but is rather partially controlled by plastic strain localisation resulting in a heterogeneous asymmetric velocity field over time. This heterogeneity is reflected in the total melt accreted over time.

We have added a short explanation to the manuscript clarifying how in places three times the reference igneous crustal thickness is accreted.

We note that in earlier studies the term “active mantle upwelling” was used to explain the voluminous magmatism at the US East Coast volcanic margin (Kelemen & Holbrook, 1995), without providing a physical explanation for what may cause the difference of the rate of mantle upwelling and plate divergence but loosely referring to either small scale convection or plume related high mantle upwelling rates. Here we provide a self-consistent explanation for this difference. Interestingly, Kelemen & Holbrook (1995) argue that excess magmatism along the US East Coast volcanic margin should be explained by high active mantle welling rates but exclude a mantle plume for several reasons: (1) lack of evidence for a hotspot track; (2) magmatism during rifting spanned at least 38 m.y., in contrast to a transient plume head model that would supposedly be characterised by a shorter timescale; (3) magma production does not exhibit a radial pattern with a decrease away from the centre of a supposed plume impact if there was. We believe that our wide rifting model provides a physical explanation for inferred active mantle upwelling and explains these discrepancies and comment briefly on this in the discussion.

Line 101 Linear correlation between margin width and total magmatic volume: this result is difficult to follow because up to here the metrics for margin width and total magmatic volume have not been clearly introduced. A brief definition should be stated in the main text when the terms are first used even if details of the processing are given in the Method section below. Also, it should be stated that the relationship is approximated by a linear trend (I presume for simplicity), despite the metrics in Fig. 4 showing a tendency to a more square-root or power-law-like relationship.

We have revised the text and include the definition of margin width and of the total melt volume in the main text and link this with Extended Data Figure 3.

We also state that the relationship is approximated by a linear trend. We have further revised the “linear correlation” to “quasi-linear correlation” to acknowledge the non-perfect linearity. One of the reasons it shows tendency of a square-root relationship is that there is slight secular cooling in our model. For models with wider margins, it takes longer time to reach breakup, which leads to slightly lower average mantle temperature and consequently slightly lower magmatic volume.

Lines 109-110 Melt window, thermal lid: this passage reads too much like jargon with terms that are not clearly defined or self-evident. The authors should carefully consider that this journal addresses a general audience.

We have added brief explanations and definition for melt window and thermal lid in the methods and to the captions of Figures 2 and 7. We also added a label of “melt window” in Fig. 2.

Line 112 A thicker thermal lid ... results in lower melt production: I stumbled over this sentence because up to now the results highlighted that wider margins produce more melt, but here it is stated that wider margins result in a thicker lid and lower melt production. How is this resolved?

We understand that this is confusing. The thermal lid reduces the amount of melt produced and accreted per unit spreading distance as the size of the melt window is limited at its top. For our models this is about 0.6 times the amount of igneous crust accreted during oceanic spreading. However, this melt comes in excess to what is produced after lithosphere breakup and adds therefore melt volume. A second point to consider is that even though the melt production is lower the thickness of igneous crust accreted to the extending continental crust above may exceed that of oceanic accretion, as the crust stays longer above the melt window. See also reply to comment to Line 92.

We have revised the manuscript and added a brief explanation of thermal lid in the methods section.

Lines 113-118 This passage is perhaps the weakest point of the manuscript. I did not understand much of how the authors arrived at their semi-analytical model, or how the different metrics referred to here are extracted from the model results. Since this analysis is the connecting point between models and observations it is crucial to the success of this study to render this as clear as possible. As it stands, a number of terms and symbols are not clearly defined in the text (melt thickness, extent of melting, d_r and d_s). The relationship between total melt volume, margin width and mature oceanic crustal thickness (related to mantle temperature, right?) is

clearly not a complicated one, indeed it would appear to be a simple linear fit to the data points in Fig. 4. However, I still don't follow what exactly this metric represents and how the authors arrived here. (After writing this comment I read on to the end of the manuscript, where these matters are quite helpfully visualised in the Extended Data Figures. I would strongly recommend to rewrite this paragraph for clarity, to consider whether some of the conceptual visualisation of this analysis could be added to main text figures, or to at least clearly refer to the Extended Data Figures if that is where the related information remains.)

We have revised the text to carefully explain how we arrive at our semi-analytical prediction law. We have greatly simplified the discussion of the semi-analytical scaling law in the main text and supplement this by a more detailed discussion in the methods section. We also updated Extended Data Fig. 4 to better illustrate how the total melt volume is dependent on mantle potential temperature and margin width.

Syn-rift magmatism is caused by the preferential removal of mantle lithosphere during depth-dependent wide rifting. The space created by removal of mantle lithosphere is proportional to margin width. Therefore, the volume of syn-rift magmatism is also proportional to margin width. The key point is how to calibrate the slope of volume-width curve. For spreading system, it is clear that oceanic crustal thickness ($V_0 = h_{oc} * W_s$, so that $h_{oc} = dV_0/dW_s$) captures melt production per increment spreading. Similarly, $h_{eff} = dV/dW$ captures melt production per increment extension. The two quantities can be linked by comparing their melt fractions during rifting and spreading (Extended Data Fig. 4). The calibration from Model II suggests that h_{eff} is approximately $0.6h_{oc}$.

Line 122 Slope of the linear relationship, h_{eff} : Up to this point it had not become clear to me that h_{eff} is the slope of that linear relationship. This should be further clarified.

We have revised the text and define h_{eff} earlier in the manuscript.

If all the melt extracted from the melting regime forms oceanic crust, the total melt volume produced during each increment of spreading equals the thickness of the oceanic crust, h_{oc} (Langmuir et al., 1992). In the case of depth-dependent wide rifting, the melt volume per unit margin width (dV/dW) equals h_{eff} . See also reply to previous comment (to Line 113-118).

Line 122 Predicts and confirms: this semi-analytical scaling law cannot predict and confirm the relationship between margin width and magma production at the same time. Normally, a model would make a prediction that may be confirmed by observations or by an independent experiment/model. Here it seems that the authors derive a simplified relationship that fits their model results. If that is correct

it would not qualify as a confirmation, only as a derivative predictive metric.

We agree with the reviewer and have revised the manuscript to use “predicts” instead of “predicts and confirms”.

The reason we use “predicts” is that the scaling law is not a “linear fitting” of model results, but derived from the characteristic behaviour of wide rifting based on the melt parameterization model. It is therefore independent and fits the model results relatively well. However, the coefficients in the relationships need to be calibrated for the particular examples, such as for instance the height of melt window during rifting and during spreading which we do using the forward models.

Line 127 It remains unclear how h_{eq} and h_{oc} are related.

h_{oc} is the predicted oceanic crustal thickness that can be inversely derived from the semi-analytical law, i.e. $h_{oc} = V^*/(0.6W+100)$. We used a separate notation for oceanic crustal thickness (h_{eq}) linked to observational data. h_{eq} can be understood as the “projected” oceanic crustal thickness converted from measured melt volume and margin width of observational data.

We have revised the manuscript to avoid defining this extra thickness h_{eq} . Instead, we directly use temperature regimes, determined by the semi-analytical law, for the classification of magmatic modes. We note that in Table 1 it is useful to compare the observed thickness of the oceanic crust h_{oc} with the projected thickness as inferred from the total magmatic volume and margin width and use a more explicit notation, \tilde{h}_{oc} , for this second quantity.

Line 151ff Total melt volume: it appears that total melt volume has units of volume per length, although that is not clearly stated anywhere. I recommend to adapt the terminology to convey that this is not simply a total volume but a total volume produced per margin width. In this passage the meaning of the metric V^* above becomes a little clearer. I think it is vital to clarify exactly what the model metric above represents and why it is meaningful for comparison with natural data when it is first introduced above.

We think that there was a misunderstanding here. It is correct that the “volume” presented in this study has a unit of $[km^2]$. This is because our models are in 2-D. Therefore the melt volume here should be understood as “total volume per margin length **along strike**”, not per margin width. This is now clearly stated in the revised manuscript.

Line 158ff Margin width: if this is the same metric as used to evaluate margin width in model results then this definition should be introduced further above where the metric is first introduced. As stated above, the authors might consider adding some

of the conceptual visualisation of these metrics to a main text figure or else should clearly refer to the Extended Data Figure with that information here.

We agree with this and have revised the manuscript to refer to Extended Data Figure 3 for margin width.

Lines 183ff Text structure: the first paragraph of the Discussion section reports the results of the data analysis on natural rifted margin and would therefore sit better in a Results section. Perhaps the authors could reconsider their section structure and form sections for Model Results, Semi-analytical Scaling Law, and Natural Rift Classification, before ending the main text on a brief Discussion.

We have followed the suggestion and include a section on Natural Rift Classification before the discussion section.

Line 187 4-8 km oceanic crust: it appears that the authors here use oceanic crustal thickness to discriminate between magmatic margin regimes. This is also what Fig. 6 confirms. On the other hand, oceanic crustal thickness is taken as proxy for mantle potential temperature. It would therefore appear that excess-, normal-, and a-magmatic regimes are determined by mantle potential temperature, not margin width. This seems to run counter to the overall conclusion of the article that margin width is the important controlling parameter, and that mantle potential temperature has secondary effects. It is possible that I have misunderstood part of the argument here but it would certainly be worth to clarify the relative roles of T_p and W in classifying ridge regimes both in the model and in natural data.

There is a misunderstanding here. We concluded that both potential temperature and margin width are primary controlling factors of the volume of magmatism.

We use temperature regimes, defined using oceanic crustal thickness as a proxy, to discriminate between magmatic modes. As illustrated in Fig. 4, each potential temperature has its corresponding semi-analytical curve with a given oceanic crustal thickness. The normal-temperature regime (yellow colour in Fig. 6a), defined by temperature range of 1280-1330 °C, roughly corresponds to oceanic crust thicknesses of 4-8 km in our models (Fig. 4). In order to better illustrate the linkage to temperatures, we additionally project model data into Fig. 6 and label their potential temperatures.

Fig. 5 This Figure is very helpful for understanding how analytical metrics are extracted from observations. A similar figure would be helpful for how metrics are extracted from model results as well.

We have added the definition of margin width in Extended Data Fig. 3, which follows the same measurement as for observations, and prefer to keep Fig. 5 as is. We note that we use the same metrics for the models and for the observations.

Line 482 Accumulated strain: does this refer to total visco-plastic strain or only plastic failure strain?

It is the accumulated visco-plastic (total) strain. We have updated the text in the revised manuscript.

Line 484 It is not made clear in the text what experimental creep parameters are used for the crustal and mantle layers. In the model setup figure only wet quartz and wet olivine are listed. Does that mean that upper and lower crust have the same rheology? If so, what else distinguishes these layers?

We now note in the methods section that the crust and mantle follow Wet Quartz and Wet Olivine flow laws respectively. The upper crust and lower crust layers are identical in terms of their properties; they have different material ids (colours) for visualization only. We have revised the text to make this clear.

Line 512 Is latent heat consumption also included in the energy equation used to solve for T? If so then it should be noted in the paragraph on the thermal model.

Latent heat consumption is included in the energy equation. This is now noted in the methods section.

Extended Data Figures are very helpful and clarify a number of questions that came up while reading the main text. I encourage the authors to consider if some of that information (particularly the visualisation of analytical metrics used to derive the semi-analytical scaling law) could be transferred to the main text. Either way, these helpful figures should be more explicitly referred to in the main text.

These figures are quite technical and we prefer to keep them as part of the Extended Data. We have, however, updated the main text and refer more explicitly to these Extended Data Figures.

Best Regards,
Tobias Keller (Tobias.Keller@glasgow.ac.uk)

References:

- Armitage, J. J. (2008). *Modelling the Controls on Melt Generation During Continental Extension and Breakup*. University of Southampton.
- Armitage, J. J., Collier, J. S., & Minshull, T. A. (2010). The importance of rift history for

- volcanic margin formation. *Nature*, 465(7300), 913–917.
<https://doi.org/10.1038/nature09063>
- Bown, J. W., & White, R. S. (1994). Variation with spreading rate of oceanic crustal thickness and geochemistry. *Earth and Planetary Science Letters*, 121(3–4), 435–449. [https://doi.org/10.1016/0012-821X\(94\)90082-5](https://doi.org/10.1016/0012-821X(94)90082-5)
- Brune, S., Heine, C., Pérez-Gussinyé, M., & Sobolev, S. V. (2014). Rift migration explains continental margin asymmetry and crustal hyper-extension. *Nature Communications*, 5, 4014. <https://doi.org/10.1038/ncomms5014>
- Clerc, C., Ringenbach, J. C., Jolivet, L., & Ballard, J. F. (2018). Rifted margins: Ductile deformation, boudinage, continentward-dipping normal faults and the role of the weak lower crust. *Gondwana Research*, 53, 20–40.
<https://doi.org/10.1016/j.gr.2017.04.030>
- Deng, H., Ren, J., Pang, X., Rey, P. F., McClay, K. R., Watkinson, I. M., et al. (2020). South China Sea documents the transition from wide continental rift to continental break up. *Nature Communications*, 11(1).
<https://doi.org/10.1038/s41467-020-18448-y>
- England, P. (1983). Constraints on extension of continental lithosphere. *Journal of Geophysical Research*, 88, 1145–1152.
- Forsyth, D. W. (1992). Geophysical Constraints on Mantle Flow and Melt Generation Beneath Mid-Ocean Ridges. In J. P. Morgan, D. K. Blackman, & J. M. Sinton (Eds.), *Mantle Flow and Melt Generation at Mid-Ocean Ridges* (pp. 1–65). Washington, D. C.: American Geophysical Union.
<https://doi.org/10.1029/GM071p0001>
- Grevemeyer, I., Ranero, C. R., & Ivandic, M. (2018). Structure of oceanic crust and serpentinitization at subduction trenches. *Geosphere*, 14(2), 395–418.
<https://doi.org/10.1130/GES01537.1>
- Holbrook, W. S., & Kelemen, P. B. (1993). Large igneous province on the US Atlantic margin and implications for magmatism during continental breakup. *Nature*, 364(6436), 433–436.
- Hopper, J. R., Mutter, J. C., Larson, R. L., & Mutter, C. Z. (1992). Magmatism and rift margin evolution: Evidence from northwest Australia. *Geology*, 20(9), 853–857.
[https://doi.org/10.1130/0091-7613\(1992\)020<0853:MARMEE>2.3.CO;2](https://doi.org/10.1130/0091-7613(1992)020<0853:MARMEE>2.3.CO;2)
- Huismans, R. S., & Beaumont, C. (2003). Symmetric and asymmetric lithospheric extension: Relative effects of frictional-plastic and viscous strain softening. *Journal of Geophysical Research: Solid Earth*, 108(B10).
<https://doi.org/10.1029/2002JB002026>
- Huismans, R. S., & Beaumont, C. (2011). Depth-dependent extension, two-stage breakup and cratonic underplating at rifted margins. *Nature*, 473(7345), 74–78.
<https://doi.org/10.1038/nature09988>
- Huismans, R. S., & Beaumont, C. (2014). Rifted continental margins: The case for depth-dependent extension. *Earth and Planetary Science Letters*, 407, 148–162.
- Jones, C. H., Wernicke, B. P., Farmer, G. L., Walker, J. D., Coleman, D. S., McKenna, L. W., & Perry, F. V. (1992). Variations across and along a major continental rift: an interdisciplinary study of the Basin and Range Province, western USA.

Tectonophys., 213, 57–96.

- Keen, C. E., & Boutilier, R. R. (2000). Interaction of rifting and hot horizontal plume sheets at volcanic margins. *Journal of Geophysical Research: Solid Earth*, 105(B6), 13375–13387. <https://doi.org/10.1029/2000jb900027>
- Kelemen, P. B., & Holbrook, W. S. (1995). Origin of thick, high-velocity igneous crust along the US east coast margin. *Journal of Geophysical Research*, 100(B6).
- Langmuir, C. H., Klein, E. M., & Plank, T. (1992). Petrological Systematics of Mid-Ocean Ridge Basalts: Constraints on Melt Generation Beneath Ocean Ridges. In J. P. Morgan, D. K. Blackman, & J. M. Sinton (Eds.), *Mantle Flow and Melt Generation at Mid-ocean Ridges* (pp. 183–280). Washington, D. C. <https://doi.org/10.1029/gm071p0183>
- McKenzie, D. (1978). Some remarks on the development of sedimentary basins. *Earth and Planetary Science Letters*, 40(1), 25–32.
- Nielsen, T. K., & Hopper, J. R. (2004). From rift to drift: Mantle melting during continental breakup. *Geochemistry, Geophysics, Geosystems*, 5(7). <https://doi.org/10.1029/2003GC000662>
- Royden, L., & Keen, C. E. (1980). Rifting process and thermal evolution of the continental margin of Eastern Canada determined from subsidence curves. *Earth and Planetary Science Letters*, 51(2), 343–361. [https://doi.org/10.1016/0012-821X\(80\)90216-2](https://doi.org/10.1016/0012-821X(80)90216-2)
- Theunissen, T., & Huisman, R. S. (2019). Long-Term Coupling and Feedback Between Tectonics and Surface Processes During Non-Volcanic Rifted Margin Formation. *Journal of Geophysical Research: Solid Earth*, 124(11), 12323–12347. <https://doi.org/10.1029/2018JB017235>

REVIEWERS' COMMENTS

Reviewer #1 (Remarks to the Author):

I am happy with the revisions made to the manuscript and have no new comments.

The only (small) issue I have is the continued use of the Bown and White 94 dataset in EDFig 2 when the authors have prepared an updated Grevemeyer et al 2018 one. As a member of the observational marine geophysics community this does make it look as if we havnt progressed much over the past 30 years!

Reviewer #2 (Remarks to the Author):

I would like to thank the authors for responding to my previous comments and suggestions in a careful and thorough manner, and find that the revised manuscript is now of sufficient clarity of argument and quality of research and presentation to warrant publication in its present form.

Tobias Keller

We appreciate the two reviewers for their final comments. Original comments are in black and our replies are in blue.

REVIEWER COMMENTS

Reviewer #1 (Remarks to the Author):

I am happy with the revisions made to the manuscript and have no new comments.

The only (small) issue I have is the continued use of the Bown and White 94 dataset in EDFig 2 when the authors have prepared an updated Grevemeyer et al 2018 one. As a member of the observational marine geophysics community this does make it look as if we havnt progressed much over the past 30 years!

We follow the reviewer's comment and have updated to use the new dataset by Grevemeyer et al. 2018 in Extended Data Fig. 2.

Reviewer #2 (Remarks to the Author):

I would like to thank the authors for responding to my previous comments and suggestions in a careful and thorough manner, and find that the revised manuscript is now of sufficient clarity of argument and quality of research and presentation to warrant publication in its present form.

Tobias Keller